

# Selected Alu methylation levels in the gastric carcinogenesis cascade

Jiraroch Meevassana[1], Chawisa Wanda Vongsuly[1],
Tanchanok Nakbua[1], Supitcha Kamolratanakul[2],
Pichaya Thitiwanichpiwong[3], Fardeela Bin-Alee[4], Somboon Keelawat[5]
and Nakarin Kitkumthorn[6]

[1] Center of Excellence in Burn and Wound Care, Faculty of Medicine, Chulalongkorn University, Bangkok, Thailand
[2] Department of Clinical Tropical Medicine, Faculty of Tropical Medicine, Mahidol University, Bangkok, Thailand
[3] Digital Pathology Center, Siriraj Piyamaharajkarun Hospital, Mahidol University, Bangkok, Thailand
[4] Faculty of Medicine, Princess of Naradhiwas University, Narathiwat, Thailand
[5] Department of Pathology, Faculty of Medicine, Chulalongkorn University, Bangkok, Thailand
[6] Department of Oral Biology, Faculty of Dentistry, Mahidol University, Bangkok, Thailand

## ABSTRACT

**Background:** Genome-wide hypomethylation, a common epigenetic change that occurs during cancer development, primarily affects repetitive elements, such as Alu repeats. Consequently, Alu repeats can be used as a surrogate marker of genomic hypomethylation.
**Methods:** In this study, we aimed to investigate the correlation between Alu methylation levels and the multistage course of gastric carcinogenesis.
**Results:** We found that the Alu methylation levels in gastric cancer tissue decreased compared with those in normal gastric tissue, with the change in methylation levels and pattern being most significant between chronic gastritis and intestinal metaplasia. Moreover, Alu methylation levels were not associated with *Helicobacter pylori* or Epstein–Barr virus infection.
**Conclusions:** Finally, our sensitivity and specificity analyses suggested that Alu methylation level can be used to distinguish gastric cancer tissue from normal tissue. Thus, Alu methylation level shows promise as biomarker for gastric cancer diagnosis.

## INTRODUCTION

Despite recent reductions in the prevalence of gastric cancer (GC) and the associated death rate, GC continues to pose considerable health concerns worldwide. Currently, GC is the fifth most commonly reported malignancy and the third leading cause of cancer-related mortality globally (*Thrift & El-Serag, 2020*). Early-stage GC is often asymptomatic or causes only non-specific symptoms; consequently, most patients present with advanced-stage cancer because of delayed diagnosis (*Xia & Aadam, 2022*; *Pasechnikov et al., 2014*). Surgery is the only available radical treatment for GC (*Song et al., 2017*).

Corresponding authors
Jiraroch Meevassana,
drjiraroch@gmail.com
Nakarin Kitkumthorn,
Nakarinkit@gmail.com

However, the curative surgical window is often missed because of the late stages of diagnosis and dissemination, contributing to the poor prognosis of GC (*Pasechnikov et al., 2014*; *Song et al., 2017*; *Kanda & Kodera, 2015*). Therefore, importance should be placed on early GC detection using effective screening approaches and the development of novel diagnostics to improve GC-related mortality.

Similar to all malignant tumors, early GC detection relies on understanding its multistep, multifactorial etiology and pathogenesis. Gastric adenocarcinomas, particularly the more prevalent intestinal type, follow a well-characterized sequence of histopathological transformations. This progression begins with normal gastric mucosa, advancing to non-atrophic chronic gastritis, followed by the development of multifocal atrophic gastritis, resulting in intestinal metaplasia, which may further evolve into gastric dysplasia, ultimately leading to the onset of gastric adenocarcinoma (*Correa & Piazuelo, 2012*). *Helicobacter pylori* infection is the predominant cause of early gastritis and the onset of chronic gastritis, which are critical stages in the advancement of gastric carcinogenesis. Multiple environmental factors are involved in this complex process (*Correa & Piazuelo, 2012*). Additionally, approximately 10% of GC cases can be attributed to Epstein–Barr virus (EBV) infection (*Naseem et al., 2018*). Although the exact mechanisms remain unclear, evidence suggests that EBV induces extensive promoter hypermethylation (*Thrift & El-Serag, 2020*; *Naseem et al., 2018*; *Padmanabhan, Ushijima & Tan, 2017*; *Palrasu et al., 2021*; *Matsusaka et al., 2014*).

In addition to the histopathological and environmental factors, various genetic and epigenetic alterations underlie the gastric carcinogenesis cascade. Genetic contribution to tumorigenesis is comparatively uncomplicated. Mutations in tumor suppressor genes (TSGs) or oncogenes cause either loss or gain of function, respectively. To date, several GC-related oncogenes and TSGs have been reported (*Kanda & Kodera, 2015*), including the oncogenes *ERBB3* and *CCND1* (*Kanda & Kodera, 2015*; *Shan et al., 2017*) and the TSG *p53*, which occurs in over 60% of GCs (*Smith et al., 2006*). The epigenetic contribution to tumorigenesis is more elaborate than genetic contribution. Epigenetics is regulated by chromatin structure, which is controlled by various processes, including methylation, acetylation, phosphorylation, ubiquitylation, nucleosome remodeling, and regulation by non-coding regulatory RNAs (*Baylin & Jones, 2016*). Epigenetic alterations can affect the expression of TSGs or oncogenes, leading to tumorigenesis (*Baylin & Jones, 2016*).

Epigenetic changes that occur during carcinogenesis include promoter hypermethylation and global hypomethylation (*You & Jones, 2012*). For example, a methyl group is added to the fifth carbon atom of the cytidine ring in a CpG dinucleotide sequence at the 5′ position. This modification plays a critical role in gene regulation and epigenetic control (*Pappalardo & Barra, 2021*). Promoter hypermethylation results in transcriptional silencing; when this alteration occurs in the promoter sequences, it contributes to carcinogenesis. On the contrary, genomic hypomethylation results in genomic or chromosomal instability (*Bae et al., 2012*). Genomic hypomethylation can occur *via* several pathways. One mechanism involves the passive loss or impaired function of DNA methyltransferase 1 (DNMT1). Alternatively, active demethylation can occur through the oxidation of methylated cytosines, catalyzed by ten-eleven translocation enzymes.

Additionally, activation-induced deaminase (AID)-mediated deamination of methylcytosine, followed by base excision repair, represents another route for genomic hypomethylation (*Pappalardo & Barra, 2021*; *Sheaffer, Elliott & Kaestner, 2016*).

A more in-depth explanation of Alu elements as epigenetic regulators and their impact on genomic instability, transcriptional regulation, and retrotransposon activity (*Ye et al., 2020*; *Patchsung et al., 2018*). Alterations in the methylation of Alu and have been shown to be associated with many cancer types, hepatoma, breast cancer, ovarian cancer, acute myeloid leukaemia, T-cell lymphoblastic leukaemia, Ewing sarcoma, and colorectal cancer (*Yüksel et al., 2016*; *Jordà et al., 2017*; *Park et al., 2014*; *Belancio, Deininger & Roy-Engel, 2009*; *Chenais, 2015*; *Yooyongsatit et al., 2013*). Alu hypomethylation may disrupt heterochromatin structure, leading to a more open chromatin conformation (*Baylin et al., 2001*; *Kim, 2019*). This loss of chromatin compaction can impair genome stability by making DNA more susceptible to damage and reducing the efficiency of DNA repair mechanisms (*Grewal & Jia, 2007*). Furthermore, reduced heterochromatin formation may result in aberrant activation of normally silenced genomic regions, increasing transcriptional noise and promoting inappropriate gene expression. These changes can ultimately contribute to genomic instability and tumorigenesis by facilitating mutations, recombination events, and disruption of normal replication and transcription regulation (*Xu, Xu & Price, 2012*; *Jakob et al., 2011*; *Chevalier et al., 2025*). Alu elements play a multifaceted role in cancer development through mechanisms involving epigenetic regulation, genomic instability, and RNA interference. Under normal physiological conditions, Alu sequences are heavily methylated to suppress their transpositional activity and maintain genomic stability (*Bhat et al., 2022*). However, in various malignancies such as gastric, colorectal, and lung cancers, global DNA hypomethylation leads to increased Alu activity (*Ye et al., 2020*). This reactivation results in insertional mutagenesis, deletions, and chromosomal rearrangements, thereby contributing to genomic instability and cancer progression (*Patchsung et al., 2018*; *Ehrlich, 2009*; *Li et al., 2025*; *Oomen et al., 2025*). Beyond their role in genome structure, Alu elements also influence gene expression through transcriptional regulation (*Jayaraman et al., 2025*). They contain transcription factor binding sites, and when positioned near oncogenes or tumor suppressor genes, they can modulate gene transcription, often promoting uncontrolled cellular proliferation (*Jordà et al., 2017*; *Chenais, 2015*; *Baylin et al., 2001*; *Grewal & Jia, 2007*). For instance, Alu sequences have been shown to affect the activity of the tumor suppressor p53 in several cancer models (*Bao et al., 2017*; *Pfeifer, 2000*; *Mangoni et al., 2025*). Furthermore, Alu elements are involved in RNA-based regulatory pathways by giving rise to non-coding RNAs, including Alu-derived microRNAs such as miR-23a. These miRNAs regulate genes implicated in cell cycle progression, apoptosis, and immune responses within cancerous tissues. Accumulation of Alu RNAs can also activate inflammatory pathways, such as NF-κB signaling, thereby contributing to the remodeling of the tumor microenvironment (*Wang et al., 2018*; *Lee & Dutta, 2009*). Due to their enrichment in gene-dense genomic regions, Alu elements serve as potential hotspots for tumorigenic mutations (*Gu et al., 2016*). Their deregulation, particularly through hypomethylation, underscores their significant role in the early stages of tumorigenesis.

Genome-wide methylation mainly affects repetitive elements such as Alu repeats, LINE-1, satellite-α (SATα), and juxtacentromeric satellite 2 sequences (SAT2) (*Ehrlich, 2009*). Alu repeats are short interspersed elements (SINEs) considered to be derived from the 7SL RNA gene (*Khitrinskaya, Stepanov & Puzyrev, 2003*). Alu elements are the most common SINEs in humans, with more than a million copies distributed throughout the genome, comprising 10% of the genetic material. They are predominantly distributed in gene-rich regions (*Khitrinskaya, Stepanov & Puzyrev, 2003*). Therefore, Alu markers can be used as representative markers of global methylation levels. Notably, diseases reportedly linked to alterations in Alu or LINE-1 methylation include hereditary diseases such as facioscapulohumeral muscular dystrophy syndrome, ataxia telangiectasia, and Aicardi-Goutières (*Pappalardo & Barra, 2021*; *Himeda & Jones, 2019*); disorders caused by the immune system including diabetes type 1, rheumatoid arthritis, and systemic lupus erythematosus (*Li et al., 2021*; *Hedrich et al., 2017*); neurologic diseases including autism spectrum disorders, Alzheimer's disease, and major depressive disorder (*Pappalardo & Barra, 2021*; *Tremblay & Jiang, 2019*; *Bollati et al., 2011*); burn scars (*Meevassana et al., 2022*); as well as cancer and aging (*Pappalardo & Barra, 2021*; *Luo, Lu & Xie, 2014*).

The aim of this study was to analyze the relationship between global DNA methylation levels and the advancing phases of gastric carcinogenesis. We investigated the temporal dynamics of global hypomethylation during gastric carcinogenesis. We examined Alu methylation levels at different stages of this process by comparing the methylation profiles across normal gastric (NS), chronic gastritis (CG), intestinal metaplasia (IM), gastric dysplasia (GD), and gastric adenocarcinoma (GA) tissues. Moreover, we evaluated whether *H. pylori* and EBV infections affected global hypomethylation during the carcinogenesis cascade. Furthermore, we determined the sensitivity of Alu methylation level in detecting early GA. We anticipate that the findings of this study will enhance our understanding of epigenetic mechanisms involved in GC pathogenesis. Moreover, we aimed to assess the potential of Alu methylation as a reliable biomarker for the early detection of GA. If proven effective, these measurements may offer a non-invasive alternative to traditional biopsy and histopathological examination in future clinical practice.

## MATERIALS AND METHODS

### Study design, sample size, and population

Formalin-fixed paraffin-embedded (FFPE) biopsied specimens were obtained from 14 patients with NS, 32 with CG, 36 with IM, 15 with GD, and 29 with GA at the Pathological Department, Faculty of Medicine, Chulalongkorn University between January 2022 and January 2023 (Table 1). Samples were collected from patients who had undergone primary endoscopic biopsy and had not taken any medications that could suppress *H. pylori* infection. A histopathological review by Somboon Keelawat of the natural killer cells was conducted to confirm the diagnosis. The exclusion criteria included limited amounts of pathological tissue, lack of clinical data, and poor DNA quality.

**Table 1 Alu methylation in gastric carcinogenesis tissue cascade.**

| | No | Sex (male, female) | Age (median, range; y) | *H. pylori* infection (positive, negative) | EBV infection (positive, negative) | Alu methylation (Avg., SD; %) |
|---|---|---|---|---|---|---|
| Normal stomach (NS) | 14 | 6, 8 | 53, 5, 18–93 | 1, 13 | 0, 14 | 43.51, 3.18 |
| Chronic gastritis (CG) | 32 | 15, 17 | 64, 31–81 | 13, 19 | 0, 32 | 44.58, 2.45 |
| Intestinal metaplasia (IM) | 36 | 15, 21 | 62, 22–85 | 9, 27 | 0, 36 | 39.09, 1.98 |
| Gastric dysplasia (GD) | 15 | 7, 8 | 72, 49–92 | 0, 15 | 0, 15 | 41.66, 4.04 |
| Gastric adenocarcinoma (GA) | 29 | 8, 21 | 68, 27–94 | 8, 21 | 4, 25 | 38.79, 1.78 |

## Ethics declaration

This study was approved by the Review Committee of the Medical Faculty of Chulalongkorn University in Bangkok, Thailand (IRB number 544/64, COA code 983/2021). All procedures were conducted according to the principles of the Declaration of Helsinki (1975) with the latest amendments from 2013. Written informed consent was obtained from the participants before enrollment. To ensure the protection of personal data, all samples and clinical information were anonymized and coded. Access to identifiable data was restricted to authorized personnel only, and all data were stored securely in protected databases.

## Microscopic analysis

The biopsy specimens were fixed in 10% neutral formalin solution. Thereafter, the specimens were embedded in paraffin and cut into 2-μm-thick sections. The sections were subsequently subjected to hematoxylin and eosin (H&E) staining for microscopic examination.

## Giemsa staining for *H. pylori*

In addition to the H&E-stained slides used for microscopic analysis, a separate slide was stained with Giemsa stain to detect *H. pylori*. Briefly, thin slices of the specimens were carefully placed on dry, clean, microscopic glass slides. After deparaffinization, the slides were hydrated with distilled water. A mixture of Giemsa staining solution, distilled water (1:20 ratio), and 12.5% methanol was used to prepare *H. pylori* solution. The sections were then incubated in the *H. pylori* solution for 15–30 min. Glass covers were placed over the slides, and the presence of *H. pylori* was determined based on the observation of curved rods. Histological assessment was performed by two pathologists (SK and NK) who were blinded to the clinical data and endoscopic findings.

## *In situ* hybridization for EBV

We assessed EBV using *in situ* hybridization (ISH) to detect EBER. The EBER-ISH procedure was conducted on 2-μm-thick FFPE histological sections. Briefly, tissue sections

were stained using an EBV probe/antibody ISH kit (Leica, Newcastle upon Tyne, UK) in conjunction with a Ventana Benchmark XT automated slide stainer (Roche, Tucson, AZ, USA). To enhance the staining process, an Ultra Vision Large Volume Detection System Anti-Polyvalent HRP (Thermo Fisher Scientific, Waltham, MA, USA) was used according to the manufacturer's instructions. Hybridization signals were visualized through a reaction with the ImmPACT™ DAB Peroxidase Substrate (Vector Laboratories, Burlingame, CA, USA). After staining, the sections were dehydrated with ethanol and xylene and subsequently mounted using suitable mounting medium.

## DNA extraction and bisulfite modification

FFPE samples were cut into 2-µm-thick sections using a microtome (Microm HM355S; Thermo Fisher Scientific, Walldorf, Germany). The sections were then deparaffinized with xylene. Genomic DNA was extracted using 10% sodium dodecyl sulfate (SDS) (Thermo Fisher Scientific, Fremont, CA, USA) lysis buffer to disrupt cell membranes, coupled with proteinase K enzymatic digestion to degrade proteins and ensure the release of high-quality nucleic acids. This process was followed by a conventional phenol/chloroform extraction method, which effectively separates DNA from proteins and other cellular contaminants. After extraction, the genomic DNA was quantitatively examined using a NanoDrop ND-1000 Spectrophotometer (NanoDrop Technologies, Wilmington, DE, USA). The integrity and suitability of the DNA for downstream applications were confirmed using spectrophotometric analysis, which allowed for an accurate assessment of the absorbance at 260/280 and 260/230 nm. This evaluation ensured minimal contamination with proteins and organic compounds.

Following DNA extraction, 500 ng of the purified genomic DNA was bisulfite-converted using an EZ DNA Methylation-Gold™ kit (Zymo Research, Irvine, CA, USA) according to the manufacturer's guidelines. Bisulfite treatment promotes the deamination of unmethylated cytosine bases, transforming them into uracil. This biochemical alteration enabled precise methylation-specific assays for subsequent analyses. The high efficiency of this conversion process is critical for accurate epigenetic studies, including DNA methylation profiling, in biomedical and genetic research. This method ensures the preservation of methylated cytosines, enabling precise downstream applications such as methylation-specific polymerase chain reaction (PCR) and bisulfite sequencing (*Patchsung et al., 2018*; *Yooyongsatit et al., 2013*; *Meevassana et al., 2022*; *Prucksakorn et al., 2025*; *Jiraboonsri et al., 2024*).

## COBRA

After bisulfite conversion of DNA, the methylation status of the CpG located within the Alu repetitive sequences was assessed *via* PCR. The amplification process began with denaturation at 95 °C for 15 min, followed by 35 cycles of denaturation at 95 °C for 45 s, annealing at 57 °C for 45 s, and extension at 72 °C for 45 s. The extension step at 72 °C for 15 min facilitated the synthesis of the PCR products. The primers utilized in this reaction included the following: forward 5′-GGYGUGGTGGTTTAYGTTTGTAA-3′ and reverse 5′-CTAACTTTTTATATTTTTAATAAAAACRAAATTTCACCA-3′. The primers were

developed using the Alu repetitive sequence obtained from GenBank (NM_031483.7). These sequences have undergone prior validation and have been effectively utilized in our investigations to ensure the precise analysis of CpG methylation (*Prucksakorn et al., 2025*; *Jiraboonsri et al., 2024*).

Examining Alu methylation patterns is important for understanding the pathogenesis of various conditions, including cancer, aging-related processes, and other genetic disorders. Such studies will contribute to elucidating how epigenetic modifications affect genomic stability and gene regulation. Following PCR amplification, the PCR products were cut using a TaqI restriction enzyme (Thermo Fisher Scientific, Waltham, MA, US), with the reaction performed under optimal conditions at 65 °C for 16 h (*Patchsung et al., 2018*; *Yooyongsatit et al., 2013*; *Meevassana et al., 2022*; *Prucksakorn et al., 2025*; *Jiraboonsri et al., 2024*). After digestion, the samples were subjected to gel electrophoresis using an 8% non-denaturing polyacrylamide gel to resolve the digested fragments based on their size. The resulting DNA fragments were visualized by staining the gel with SYBR Green (Lonza Group, Ltd., Basel, Switzerland) for 30 min; SYBR Green binds to nucleic acids and enable fluorescence-based detection. The DNA band intensity was quantitatively measured using Strom840 imaging and ImageQuant NT software (Amersham; Cytiva, Marlborough, MA, USA) to assess the degree of methylation based on the band patterns. For quality assurance and reliability of the agarose gel electrophoresis and subsequent band intensity analysis, DNA extracted from HeLa cells was used as a positive control to confirm the accuracy of agarose gel electrophoresis and band intensity quantification. As a negative control, a full reaction mixture with all reagents except DNA, to ensure there was no contamination or non-specific enzymatic activity during the assay. COBRA assays were performed in duplicate to ensure technical consistency, following the approach used in our previous study (*Meevassana et al., 2022*; *Chaiwongkot et al., 2022*; *Meevassana et al., 2022*).

## Alu methylation analysis

The COBRA technique assesses DNA methylation patterns by analyzing band lengths associated with different CpG locus configurations. These patterns were categorized by methylation status at the two CpG sites within the Alu elements. Specifically, Alu sequences can exhibit the following patterns: two unmethylated CpG sites (denoted as uCuC), which produce a band of 133 base pairs (bp); two fully methylated CpGs (mCmC), resulting in bands of 58 and 32 bp; 1 uCmC, forming a 75 bp band; and 1 mCuC, yielding a 90-bp band. This method is particularly useful for distinguishing the methylation status across multiple CpG sites, providing valuable insights into epigenetic regulation within repetitive elements such as Alu sequences. Variation in band length correlates directly with specific methylation patterns, which is critical for understanding the functional consequences of DNA methylation in genomic regulation and disease. To quantify the intensity of each electrophoretic band, the intensity measured for each band was normalized by dividing it with the corresponding band length. Specific calculations for each band were as follows: (A) 133 bp divided by 133; (B) 58 bp divided by 58; (C) 75 bp divided by 73; (D) 90 bp divided by 90; and (E) 43 bp divided by 41. Subsequently, the methylation status of the Alu loci was determined using several formulas and expressed as

percentage. The overall Alu methylation level, denoted as %mC, was calculated using the following formula: $\%mC = 100 \times (B + E)/(2A + B + C + D + E)$. Further classification of the methylation status at specific loci involved determining the percentage of loci that were fully methylated at both CpG sites (%mCmC), hemimethylated at one CpG site (%uCmC and %mCuC), or fully unmethylated (%uCuC). To calculate the percentage contribution of each component within a system, we followed a similar approach for each component by determining its value relative to the total sum of all components (A, C, D, and F).

The percentage of mCmC was determined by dividing the value of component F by the sum of all components (A, C, D, and F) and then converting this fraction into a percentage. The percentage of mCmC was calculated as $(F/(A + C + D + F)) \times 100$.

For uCmC, we calculated the percentage by dividing the value of component C by the total sum of all components and then converting that value into a percentage. The percentage of uCmC was determined as follows: $(C/(A + C + D + F)) \times 100$.

To determine the percentage of mCuC, the value of component D was divided by the total of all components and converted into a percentage. The percentage of mCuC was computed as $(D/(A + C + D + F)) \times 100$.

Finally, the percentage of uCuC was calculated by dividing the value of component A with the total sum of all components and converting the result into a percentage. The percentage of uCuC is calculated as $(A/(A + C + D + F)) \times 100$.

Each step allowed the expression of the proportion of a specific component as a percentage of the total, ensuring that the contribution of each component was accurately represented in relation to the entire system.

These detailed calculations enabled an in-depth analysis of the epigenetic status of Alu repetitive elements, which serve as important markers for genome-wide methylation profiling in biomedical and genetic research.

### Statistical analysis

In this study, we examined global Alu methylation levels and their specific patterns across NS, CG, IM, GD, and GA cohorts. The Kruskal–Wallis test was used to assess differences between groups. Pairwise comparisons using Tukey's *post-hoc* analysis were used to further investigate intergroup differences. The multivariate regression analysis was used to confirm the independent association between Alu methylation and pathological stages. The diagnostic performance of global Alu methylation was analyzed using ROC curve analysis, AUC, specificity, and sensitivity. Statistical analyses were performed using SPSS software (version 25.0; SPSS Inc., Chicago, IL, USA). Statistical significance was considered at $p < 0.05$.

## RESULTS

### Alu methylation levels along the carcinogenesis cascade

The Alu methylation levels (mC) showed a significant reduction ($p < 0.0001$) from NS and CG to IM and GA (Table 1 and Fig. 1A). Differences in the mC pattern of Alu methylation were observed between CG and IM ($p < 0.0001$), CG and GA ($p < 0.0001$), NS and IM

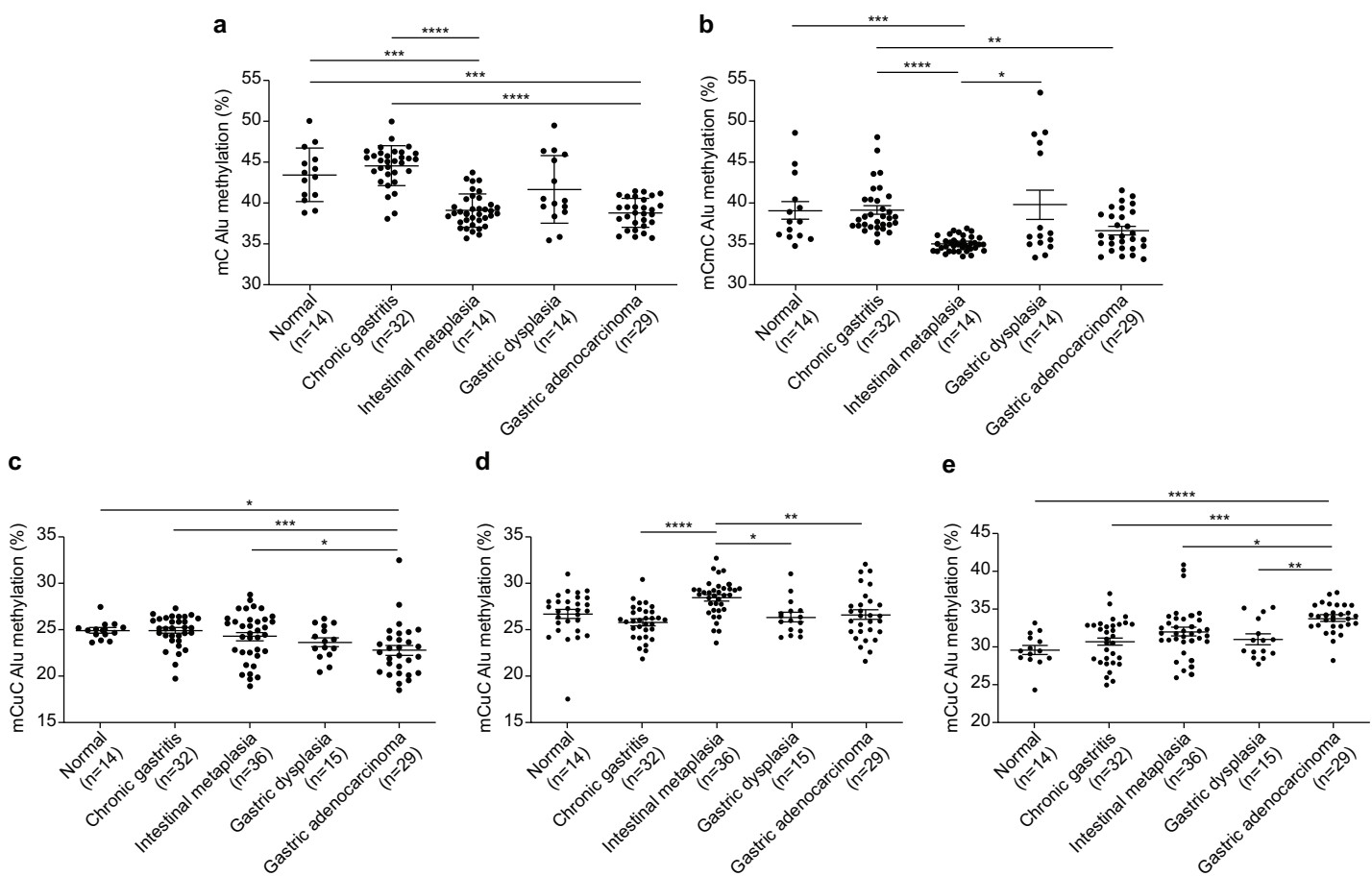

**Figure 1** **Alu methylation patterns across sample groups.** (A) The overall Alu methylation (mC) decreased significantly from NS and CG to IM and GA ($p < 0.0001$) for these comparisons. (B) Significant differences in mCmC Alu methylation were observed between CG and IM ($p < 0.0001$), NS and IM ($p = 0.0002$), CG and GA ($p = 0.0058$), and IM and GD ($p = 0.0216$). (C) The mCuC methylation pattern also significantly differed between CG and IM ($p < 0.0001$), NS and GA ($p = 0.0348$), CG and GA ($p = 0.0008$), and IM and GA ($p = 0.0258$). (D) A notable increase in uCmC Alu methylation was observed in IM compared with that in CG ($p < 0.0001$), GD ($p = 0.0100$), and GA ($p = 0.0029$). (E) In contrast to other patterns, the frequency of uCuC Alu methylation progressively increased from NS to CG, GD, and GA, with differences found between NS and GA ($p < 0.0001$), CG and GA ($p = 0.0002$), IM and GA ($p = 0.0134$), and GD and GA ($p = 0.0085$). NS, normal gastric tissue; CG, chronic gastritis; IM, intestinal metaplasia; GD, gastric dysplasia; GA, gastric adenocarcinoma. $^*p < 0.05$, $^{**}p < 0.01$, $^{***}p < 0.001$ and $^{****}p < 0.0001$.

($p = 0.0007$), and NS and GA ($p = 0.007$). Detailed frequency data are listed in Table S1. As shown in Fig. 1B, the mCmC pattern of Alu methylation was compared between CG and IM ($p < 0.0001$), CG and GA ($p = 0.0058$), NS and IM ($p = 0.0002$), and IM and GD ($p = 0.0216$).

The mCuC pattern of Alu methylation showed significant differences among the CG:IM ($p < 0.0001$), NS:GA ($p = 0.0348$), CG:GA ($p = 0.0008$), and IM:GA ($p = 0.0258$) groups (Fig. 1C). Moreover, the uCmC pattern of Alu methylation considerably increased in the IM group. The uCmC pattern of Alu methylation also significantly differed among the CG:IM ($p < 0.0001$), IM:GD ($p = 0.0100$), and IM:GA ($p = 0.0029$) groups, as shown in Fig. 1D.
In contrast to the other patterns, the uCuC pattern of Alu methylation increased from NS to CG, GD, and GA. Significant differences were found among the NS:GA ($p < 0.0001$), CG:GA ($p = 0.0002$), IM:GA ($p = 0.0134$), and GD:GA ($p = 0.0085$) groups (Fig. 1E).

## The influence of sex and age on Alu methylation across different stages of gastric carcinogenesis

We performed a multivariate regression analysis adjusting for age and sex to assess their potential influence on the percentage of Alu methylation across different stages of gastric carcinogenesis. The results showed that neither sex nor age was significantly associated with the percentage of Alu methylation in normal stomach, chronic gastritis, intestinal metaplasia, dysplasia, or adenocarcinoma ($p > 0.05$ for all comparisons). However, the percentage of mCmC methylation in adenocarcinoma was significantly associated with sex (Adjusted β = −4.09, 95% CI [−8.16 to −0.022], $p = 0.049$), and uCuC methylation was significantly associated with age (Adjusted β = −0.060, 95% CI [−0.10 to −0.017], $p = 0.008$). All statistical results are shown in Tables S2–S6.

## Alu methylation and the influence of *H. pylori* and EBV infections

Based on Giemsa staining, a significant prevalence of *H. pylori* was observed in the CG group (40.63%), followed by the GA (27.59%), IM (25.00%), and NS (7.14%) groups. Notably, all 15 cases of GD showed absence of *H. pylori*. Based on EBV-encoded small RNA (EBER) analysis results, positive EBER staining was detected in 4 out of 25 cases (16%) of GA. Conversely, all remaining lesions and normal stomachs exhibited negative EBER staining. The results are summarized in Table 1.

The Alu methylation levels in CG samples were comparable between *H. pylori*-negative and -positive samples, with percentages of 44.73 ± 2.88% and 44.36 ± 1.87%, respectively (Table S1). However, in IM samples, the Alu methylation levels increased in *H. pylori*-negative samples compared with those in *H. pylori*-positive samples, although the difference was not significant (39.31 ± 2.04% and 38.51 ± 1.79%, respectively, Fig. 2A). Similarly, in GA samples, the methylation levels in *H. pylori*-negative and -positive samples were 38.76 ± 1.88% and 38.88 ± 1.58%, respectively (Fig. 2B).

In relation to EBV infection, individuals with EBER-positive GA exhibited a slightly higher level of Alu methylation than those with EBER-negative GA. However, this difference was not significant (39.24 ± 1.49% and 38.72 ± 1.88%, respectively, as shown in Fig. 2C).

## Alu methylation levels as an additional marker for early GA detection

We assessed the efficacy of Alu methylation in detecting GA in biopsy lesions. In some cases, the gastric biopsy specimen was distorted or contained a small number of tumor cells. To address this issue, we carefully selected the most significant data points (NS + CG/GA, $p < 0.0001$; Fig. 3A) to develop a test. To determine the optimal cut-off value, we employed receiver operating characteristic (ROC) curve analysis and subsequently calculated the sensitivity, specificity, and area under the curve (AUC). As shown in Fig. 3B, the AUC of Alu methylation levels yielded a maximum value of 0.9344.

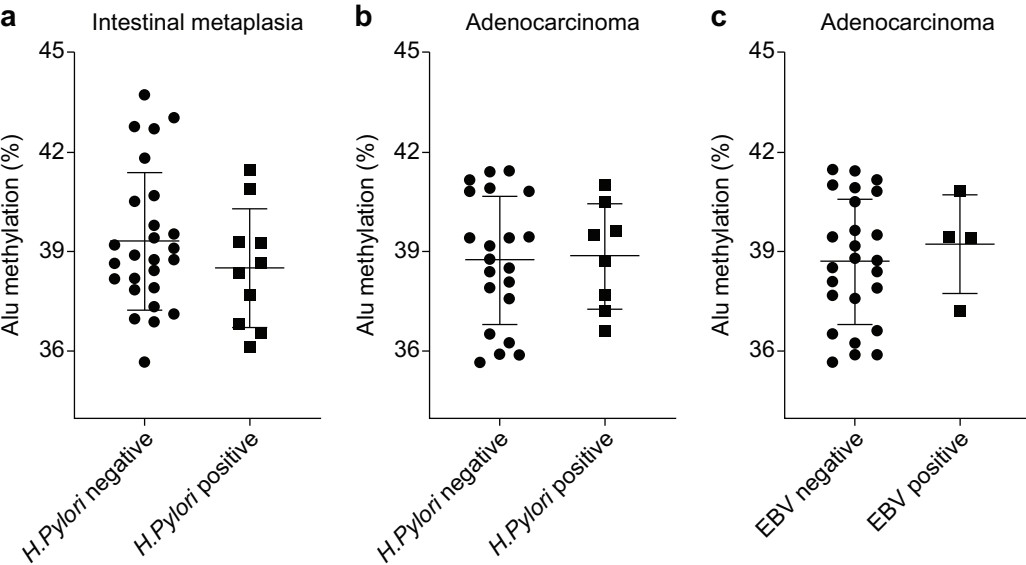

**Figure 2 Alu methylation levels in relation to *Helicobacter pylori* and Epstein–Barr virus (EBV) infections.** (A) Alu methylation in CG samples was similar between *H. pylori*-negative and -positive cases (44.73 ± 2.88% *vs.* 44.36 ± 1.87%, respectively), whereas, in IM samples, *H. pylori*-negative cases showed higher methylation levels than positive cases, though not statistically significant (39.31 ± 2.04% *vs.* 38.51 ± 1.79%). (B) In GA samples, Alu methylation between *H. pylori*-negative (38.76 ± 1.88%) and -positive (38.88 ± 1.58%) cases did not significantly differ. (C) EBV-encoded small RNA (EBER)-positive GA cases exhibited slightly higher Alu methylation levels than EBER-negative cases (39.24 ± 1.49% and 38.72 ± 1.88%, respectively), but the difference was not significant. NS, normal gastric tissue; CG, chronic gastritis; IM, intestinal metaplasia; GD, gastric dysplasia; GA, gastric adenocarcinoma.

Using methylation below 40.98% as the cut-off point, the analysis revealed 86.21% and 86.96% sensitivity and specificity, respectively.

## DISCUSSION

Carcinogenesis is driven by both genetic and epigenetic alterations, and genome-wide hypomethylation is a common epigenetic change in cancer. Genome-wide hypomethylation mainly affects repetitive DNA sequences such as Alu repeats, LINE-1, SATα, and SAT2. Therefore, repetitive elements may serve as representative genomic hypomethylation markers (*Yang et al., 2004*). Our study demonstrates a correlation between Alu methylation levels and GC progression.

The gradual reduction in Alu methylation observed from NS through CG, IM, GD, and GA indicates a progressive epigenetic modification associated with GC pathogenesis. The substantial differences in Alu methylation level between CG and both IM and GA suggest that this alteration is integral to the transformation from a chronic inflammatory state to a precancerous state and, eventually, malignancy. Furthermore, the differences among NS, IM, and GA indicate that epigenetic shifts start with disease progression, potentially before histological changes become apparent. These results indicate that Alu methylation can be used as a biomarker for diagnosing, investigating, and monitoring gastric disease progression, offering potential insights into disease pathogenesis and therapeutic

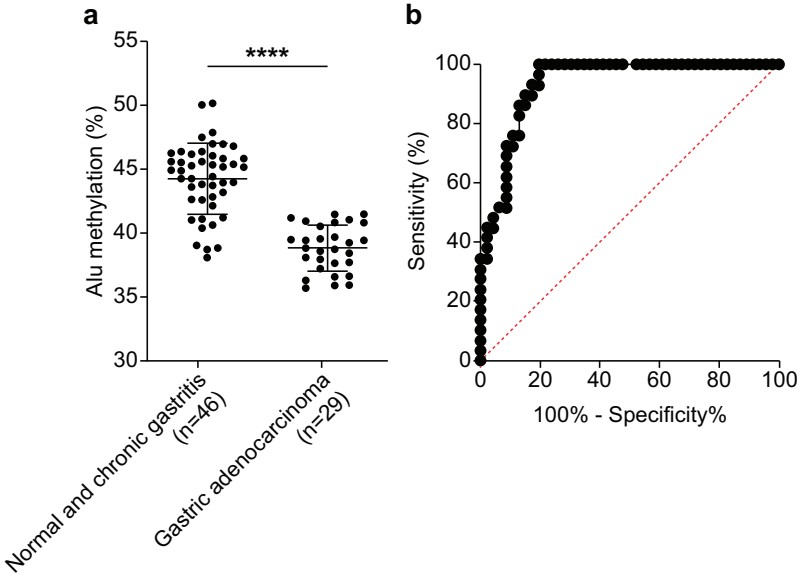

**Figure 3 Diagnostic efficacy of Alu methylation levels in detecting GA from biopsy lesions.** (A) Alu methylation levels were significantly reduced in gastric adenocarcinoma (GA; $n$ = 29) compared to non-cancerous tissues, including normal gastric tissue and chronic gastritis (NS + CG; $n$ = 46), with a highly significant difference ($p$ < 0.0001). These data suggest that Alu hypomethylation is associated with malignant transformation in gastric tissue. (B) Receiver operating characteristic (ROC) analysis demonstrated that Alu methylation has strong diagnostic potential for gastric adenocarcinoma (GA), with an AUC of 0.9344. Using a cut-off value of 40.98%, the assay achieved 86.21% sensitivity and 86.96% specificity. These findings highlight Alu methylation as a promising biomarker for the accurate detection of GA, even in biopsy samples with limited tumor content. ****$p$ < 0.0001.

strategies. Therefore, in-depth research is required to explore the methylation process and its contribution to GC development. Dual assessment of both methylation levels and patterns, acknowledging the heterogeneous nature of Alu methylation changes. Rather than using conventional pyrosequencing, we employed the COBRA-Alu method. Although COBRA analyzes fewer CpG sites than pyrosequencing, it offers comparable accuracy in determining methylation percentages (*Jintaridth & Mutirangura, 2010*; *Pobsook et al., 2011*). Additionally, COBRA-Alu yields valuable insights by distinguishing between fully methylated, fully unmethylated, and two distinct partially methylated patterns. This enables a more nuanced analysis of methylation status, offering advantages over pyrosequencing, which is unable to differentiate among mCmC, uCmC, mCuC, and uCuC configurations (*Jeddi et al., 2024*). For the Alu methylation patterns (mCmC, uCmC, and mCuC), the most pronounced variation was observed between CG and intestinal IM.

Global DNA hypomethylation, particularly within repetitive elements such as Alu and LINE-1, is a hallmark of various malignancies, including gastric cancer (*Gezer et al., 2024*). In our study, Alu hypomethylation was significantly associated with the progression of gastric lesions, supporting its role as an early epigenetic event in gastric carcinogenesis. Hypomethylation leads to derepression of normally silenced retrotransposable elements, such as Alu and LINE-1, which are abundant in gene-rich

regions. This can result in insertional mutagenesis, recombination events, and chromosomal instability—mechanisms that contribute to the disruption of genomic integrity and facilitate malignant transformation (*Xiang et al., 2010*; *Besselink et al., 2023*). Furthermore, global hypomethylation has been implicated in the dysregulation of tumor suppressor genes (TSGs) through indirect mechanisms. Although TSG silencing is more commonly attributed to promoter hypermethylation, hypomethylation-induced genomic instability may lead to structural alterations, mutations, or deletions in TSG loci such as *TP53* and *CDKN2A*, other gastric cancer-related tumor suppressor genes such as *CDH1*, *RASSF1A*, and *MLH1* impairing their tumor-suppressive functions (*Heydari et al., 2024*). Additionally, activation of transposable elements can interfere with TSG expression or disrupt regulatory networks essential for cell cycle control and apoptosis (*Gao et al., 2012*; *Hou et al., 2010*). Our finding that GA had lower overall methylation than NS is consistent with the findings of other GC studies (*Xiang et al., 2010*; *Leodolter et al., 2015*; *Park et al., 2009*). These findings in GC are consistent with those in various other cancer types (*Pappalardo & Barra, 2021*; *Ehrlich, 2009*; *Luo, Lu & Xie, 2014*), supporting the consensus that genomic hypomethylation is a key event in cancer, irrespective of the tissue type. Although various factors have been suggested to be involved, the mechanism of the genome-wide process remains poorly understood. Studies in mouse models have shown that mutations in *DNMT1* cause genome-wide DNA hypomethylation, which leads to tumor formation in mice (*Gaudet et al., 2003*). Similarly, genome-wide methylation could be caused by the dysfunction of enzymes in the one-carbon metabolic pathway (*Pogribny & Beland, 2009*). Genome-wide hypomethylation could lead to the loss of imprinting and, subsequently, the potential dysregulation of oncogenes and TSGs in those loci (*Holm et al., 2005*). Hypomethylation of retrotransposons such as Alu and LINE-1 may also cause their activation and transposition, leading to genomic instability (*Pogribny & Beland, 2009*).

Beyond the established causal relationship between genomic hypomethylation and cancer, we aimed to determine whether genomic hypomethylation occurs progressively or abruptly in the gastric carcinogenesis cascade. A few studies have examined this change from a chronological perspective (*Bae et al., 2012*). Our findings align with those of *Bae et al. (2012)*, who also demonstrated a marked reduction in Alu methylation levels during the transition from CG to IM. Additionally, *Bae et al. (2012)* reported that LINE-1 methylation levels significantly differed between IM and GA, suggesting that LINE-1 and Alu hypomethylation occurs at different stages of carcinogenesis (*Bae et al., 2012*). However, *Park et al. (2009)* reported that the Alu methylation levels were similar in CG, IM, and GA but significantly decreased in the GC stage. The same authors reported that the methylation levels of other repetitive elements, LINE-1 and SAT2, decreased progressively from CG to IM to GA (*Park et al., 2009*). In conclusion, we propose that genome-wide DNA hypomethylation is associated with the early stages of carcinogenesis, with the results of our study suggesting that the transition from CG to IM is associated with the most significant methylation change. As hypomethylation occurs early in the gastric carcinogenesis cascade, it is a promising biomarker for the detection of precancerous lesions. Correspondingly, we conducted ROC curve analysis based on our sample data, which showed that the sensitivity and specificity reached 86.21% and 86.96%,

respectively. Nevertheless, the discrepancy in the findings of different studies highlights the need for further research on the association between gastric carcinogenesis stage and hypomethylation in each repetitive element including Alu, LINE-1, SATα, and/or SAT2. Our findings, in line with previous studies, highlight that Alu hypomethylation is not only a marker of genomic instability but also a potential contributor to the early stages of tumorigenesis. Given its detectability in both tissue and blood-based assays, Alu hypomethylation may serve as a valuable biomarker for early detection and risk assessment of gastric cancer. Further mechanistic studies are warranted to delineate the exact pathways by which hypomethylation promotes tumor development and to evaluate its potential utility in clinical settings (*Xiang et al., 2010*).

Notably, our results showed that infection with *H. pylori* is not correlated with Alu hypomethylation. Our results corroborate those of *Bae et al. (2012)*, *Yoshida et al. (2011)*. Additionally, *Park et al. (2009)* found that while gene-specific methylation significantly differed in CG between *H. pylori*-positive and negative samples, genome-wide methylation did not. However, *Leodolter et al. (2015)*, *Maekita et al. (2006)*, *Shin et al. (2011)* reported that methylation levels differed significantly between *H. pylori*-positive and -negative samples. However, the discrepancy in the results may be due to the different techniques used to estimate global gene methylation. Nevertheless, conflicting findings regarding *H. pylori* infection highlight the complexity of cancer pathogenesis and the interplay among various factors, such as infection. Similarly, EBV infection was not associated with Alu hypomethylation in our study. EBV infection has been associated with significant promoter hypermethylation in other studies (*Thrift & El-Serag, 2020*; *Naseem et al., 2018*; *Padmanabhan, Ushijima & Tan, 2017*; *Palrasu et al., 2021*; *Matsusaka et al., 2014*); however, the correlation between EBV infection and global methylation remains unclear. Although previous studies have implicated *Helicobacter pylori* (*H. pylori*) and Epstein–Barr virus (EBV) infections in modulating DNA methylation patterns, our analysis revealed no significant differences in Alu methylation levels between infection-positive and -negative gastric tissues. Chronic *H. pylori* infection has been shown to induce inflammation-driven epigenetic reprogramming, leading to both global hypomethylation and gene-specific promoter hypermethylation (*Wang et al., 2024*). For instance, *Leodolter et al. (2015)* reported increased Alu hypomethylation in *H. pylori*-positive gastric mucosae, a finding not replicated in our study, possibly due to differences in bacterial strain virulence, host genetics, or timing of infection assessment. Conversely, *Park et al. (2009)* found no significant differences in global methylation, aligning with our observations and suggesting that *H. pylori*'s effects may be confined to specific loci rather than repetitive elements. Similarly, EBV-associated gastric cancer (EBVaGC) displays a unique CpG island hypermethylation phenotype, driven by latent viral proteins such as LMP2A that upregulate DNMT1 and DNMT3B expression, leading to the silencing of tumor suppressor genes including *CDKN2A*, *PTEN*, and *MLH1* (*Naseem et al., 2018*) Despite these well-characterized promoter-level changes, our study detected no significant difference in Alu methylation, suggesting that EBV's epigenetic effects may be selective and not extend to repetitive elements (*Matsusaka et al., 2014*). Variations in findings across studies may stem from differences in methodology, sample size, disease stage, host

epigenetic machinery, and viral strain heterogeneity (*Zeng et al., 2022*). Notably, some reports have proposed that co-infection with *H. pylori* and EBV may synergistically exacerbate mucosal inflammation and facilitate viral entry and persistence, further contributing to epigenetic field cancerization (*Padmanabhan, Ushijima & Tan, 2017*; *Palrasu et al., 2021*). Therefore, while our data do not support a role for global Alu hypomethylation in relation to infection status, accumulating evidence indicates that *H. pylori* and EBV exert complex, context-dependent effects on gastric DNA methylation that merit further study using diverse methylation targets and larger, multi-institutional cohorts. The results of our study improve our understanding of the pathophysiological mechanisms underlying gastric carcinogenesis induced by *H. pylori* and EBV infections.

This study has some limitations. First, we used Alu as the only marker of global hypomethylation. However, as discussed earlier, LINE-1 and Alu hypomethylation does not occur in the same stage of carcinogenesis (*Bae et al., 2012*; *Park et al., 2009*). This may be because the mechanisms regulating the methylation of repetitive DNA sequences differ (*Yang et al., 2004*). Therefore, Alu methylation may not be a perfect representative of overall genome-wide methylation. The lack of subclassification of Alu elements such as Alu J, Alu S, and Alu Y, which can display distinct methylation patterns, could also limit data interpretation. Second, we did not conduct laser capture microdissection; therefore, the samples may have contained normal cells. This implies that Alu methylation results may depend on the proportion of cancer cells in the sample. Third, the sample size in our study was relatively small, especially in certain subgroups, which may limit the statistical power and generalizability of our findings. To address this limitation and validate our results, future studies should include larger cohorts from multiple institutions or integrate publicly available datas. Multi-center studies with well-characterized samples will be essential to confirm the diagnostic value of Alu hypomethylation and its utility as a biomarker in diverse populations. This study was based on formalin-fixed, paraffin-embedded (FFPE) tissue samples, which limited the ability to evaluate dynamic temporal changes in Alu methylation across successive stages of gastric carcinogenesis within the same individuals. Consequently, the progression and potential reversibility of epigenetic alterations could not be fully assessed. To address this limitation, future studies should incorporate high-resolution methylation profiling techniques—such as bisulfite sequencing PCR (BSP), quantitative methylation-specific PCR (qMSP), or MassARRAY— to enable more precise and comprehensive analysis. We propose that future investigations incorporate DNMT1 immunohistochemistry or other molecular approaches to better elucidate the role of DNA methyltransferase activity in the regulation of repetitive element methylation during gastric tumorigenesis. Additionally, integrating gene-specific methylation and expression data will be essential to clarify the interplay between Alu hypomethylation and the regulation of tumor suppressor genes in gastric cancer. Moreover, this study did not examine the potential involvement of DNA demethylases (*e.g.*, the TET family) in regulating Alu methylation, which limits our understanding of the reversibility of epigenetic alterations in gastric cancer. Another important limitation is the lack of external validation of our ROC analysis and the absence of direct comparison with established clinical biomarkers such as CEA and CA19-9. Future studies should validate

Alu methylation findings using independent cohorts—through public databases or collaborative sample collections—and assess its diagnostic sensitivity and specificity relative to conventional biomarkers to better define its clinical utility.

## CONCLUSIONS

In conclusion, our study showed that Alu methylation levels were lower in GC tissues than in normal tissues. In particular, the change in methylation levels was most significant from CG to IM. However, Alu methylation levels were comparable among IM, GD, and GA. Moreover, Alu methylation levels were not associated with *H. pylori* or EBV infection. Our sensitivity and specificity analyses also suggest that Alu methylation levels can be used to distinguish GC tissues from normal tissue; thus, it is a promising potential biomarker for GC diagnosis. Future studies should delve into additional repeated sequences and alternative methods of methylation detection for comparative analyses.

## ACKNOWLEDGEMENTS

JM and SK would like to express our deepest gratitude to our beloved children, Suparoch Meevassana (AJ) and MUGI, whose presence brings profound strength and meaning to our lives. Their unconditional love and innocence have sustained us through the most challenging moments. We are especially thankful for their quiet support during countless late nights of research and writing, and for being a constant source of inspiration. Their very existence is a beautiful reminder of what truly matters and motivates us to strive for excellence in all we do.

### Funding
The authors received no funding for this work.

### Competing Interests
The authors declare that they have no competing interests.

### Author Contributions
- Jiraroch Meevassana conceived and designed the experiments, performed the experiments, analyzed the data, prepared figures and/or tables, authored or reviewed drafts of the article, and approved the final draft.
- Chawisa Wanda Vongsuly analyzed the data, prepared figures and/or tables, and approved the final draft.
- Tanchanok Nakbua conceived and designed the experiments, analyzed the data, prepared figures and/or tables, and approved the final draft.
- Supitcha Kamolratanakul conceived and designed the experiments, performed the experiments, prepared figures and/or tables, and approved the final draft.
- Pichaya Thitiwanichpiwong performed the experiments, authored or reviewed drafts of the article, and approved the final draft.

- Fardeela Bin-Alee conceived and designed the experiments, performed the experiments, prepared figures and/or tables, authored or reviewed drafts of the article, and approved the final draft.
- Somboon Keelawat conceived and designed the experiments, performed the experiments, authored or reviewed drafts of the article, and approved the final draft.
- Nakarin Kitkumthorn conceived and designed the experiments, performed the experiments, analyzed the data, prepared figures and/or tables, authored or reviewed drafts of the article, and approved the final draft.

## Human Ethics

The following information was supplied relating to ethical approvals (*i.e.*, approving body and any reference numbers):

Review Committee of the Medical Faculty of Chulalongkorn University in Bangkok, Thailand.

## Data Availability

The raw measurements are available in the Supplemental File.

## Supplemental Information

Supplemental information for this article can be found online at http://dx.doi.org/10.7717/peerj.19485#supplemental-information.

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
