# Peer review of "Selected Alu methylation levels in the gastric carcinogenesis cascade"

_PeerJ, doi:10.7717/peerj.19485_

## Round 0.1 · original submission · Major Revisions

The editor has a major concern on the method. Indeed, the method used to assess DNA methylation of the selected ALU element is not clear. In general extracted DNA is treated with bisulfite prior any additional step such as PCR. Please instead of referring to validation of the method in prior publications, where the validation is not determined by the use of appropriate controls, provide proof of principle that the method used is really able to give reliable information. This can be done by applying it on commercial available methylated vs unmethylated DNA.
In addition, please check that if true that 500 micrograms of DNA were used. It is more likely that 500 ng were extracted from FFPE slides.

Furthermore, it would be appropriate to change the title to “Selected Alu methylation levels in the gastric carcinogenesis cascade”.

Several important issues were raised by the reviewers, including small size of the groups and confounding factors not taken into account. It is important to address those issues as well.

Reviewer 1 ·

Basic reporting

1. Language Clarity: Some sentences in the manuscript are overly long or complex, which may hinder understanding. For example, in the abstract, "Alu methylation level is a promising potential biomarker for gastric cancer diagnosis." can be simplified to "Alu methylation level shows promise as a biomarker for gastric cancer diagnosis."
2. Background Information: While the importance of Alu elements in cancer is mentioned, the molecular mechanisms underlying their roles are insufficiently detailed.
3. Figures 2 and 3 have insufficient legends that fail to clearly explain the data.

Experimental design

1. The sample sizes for each group (e.g., 14 normal, 29 cancer tissues) are relatively small, potentially affecting the statistical power.
2. Limited Subgroup Analysis:** While different stages of gastric cancer were analyzed, confounding factors like gender and age were not thoroughly examined.

Validity of the findings

1. Mechanistic Explanation: While the correlation between Alu methylation and gastric cancer progression is established, the underlying mechanisms remain unexplored. So, discuss how Alu methylation may regulate oncogene or tumor suppressor gene expression, or reference relevant studies to provide insights.
2. Infection and Methylation Relationship: The analysis of Helicobacter pylori and Epstein-Barr virus infection shows no significant differences in methylation levels, but the biological reasons are not explored.Suggestion that expand on how infections might influence genome-wide methylation patterns and discuss the discrepancies with prior studies.

Additional comments

1. Ethics Statement: Although adherence to the Helsinki Declaration is mentioned, patient information protection measures are not described in detail.
2. Outdated References: Some cited studies on methylation mechanisms are relatively old. For instance, more recent reviews could be included.Suggestion that update citations with studies published after 2020, particularly on early diagnosis applications in gastric cancer.

Reviewer 2 ·

Basic reporting

This study investigates the dynamic changes of Alu methylation during the multistage progression of gastric carcinogenesis and its diagnostic potential. The topic is clinically significant, with a well-structured design, appropriate sample size, and clear methodological descriptions. The conclusions provide novel insights into epigenetic mechanisms and early diagnosis of gastric cancer (GC). However, the depth of analysis in certain areas is insufficient, and additional critical data are needed to strengthen the reliability of the conclusions.

Experimental design

1. Sample and Statistical Analysis
Issue: Small sample size in the GD group (n=15) may reduce statistical power; confounding factors (e.g., age, sex, smoking) were not controlled.
Recommendations:
Expand the GD cohort or explicitly acknowledge limitations (e.g., single-center recruitment challenges).
Perform multivariate regression analysis (adjusting for age/sex) to confirm the independent association between Alu methylation and pathological stages.
2. Methodological Details
Issue: Lack of information on COBRA reproducibility (e.g., inter-batch variability) and positive/negative controls; absence of laser microdissection to exclude normal cell contamination.
Recommendations:
Provide reproducibility data (e.g., coefficients of variation from triplicate experiments).
Discuss potential dilution effects from non-purified tumor cells or validate findings using laser-captured samples.
3. Mechanistic Insights
Issue: Limited exploration of the biological consequences of Alu hypomethylation (e.g., genomic instability) or links to key regulators (e.g., DNMT1 expression).
Recommendations:
Supplement with DNMT1 immunohistochemistry to correlate its expression with Alu methylation levels.
Discuss potential retrotransposon activation (e.g., LINE-1) driven by Alu hypomethylation and its oncogenic implications.
4. Diagnostic Validation
Issue: ROC analysis lacks external validation; no comparison with established biomarkers (e.g., CEA, CA19-9).
Recommendations:
Validate findings in an independent cohort (e.g., public databases or collaborative samples).
Compare sensitivity/specificity of Alu methylation against conventional biomarkers to highlight clinical utility.

Validity of the findings

Terminology: Standardize "Epstein3Barr virus" to "Epstein-Barr virus" throughout the text.
References: Formatting errors in citations (e.g., non-standard journal abbreviations in Refs. 37–38).
Ethics Statement: Explicitly state that informed consent was obtained and include the ethics approval code (only IRB number mentioned).

Reviewer 3 ·

Basic reporting

The study investigates changes in Alu methylation levels during gastric carcinogenesis, revealing a significant decrease in gastric cancer tissues, with the most pronounced change occurring from chronic gastritis (CG) to intestinal metaplasia (IM). It also examines the impact of Helicobacter pylori (H. pylori) and Epstein-Barr virus (EBV) infections, finding no significant correlation between Alu methylation levels and these infections, and proposes its potential as a biomarker for gastric cancer diagnosis.

The study includes only 14 cases of normal gastric tissue (NS), 32 cases of chronic gastritis (CG), 36 cases of intestinal metaplasia (IM), 15 cases of gastric dysplasia (GD), and 29 cases of gastric cancer (GA), resulting in a small sample size, especially in the gastric dysplasia (GD) group, with only 15 cases, leading to significant sample bias.

The study conducts methylation analysis based solely on fixed tissue samples (FFPE tissue), whereas gastric carcinogenesis is a dynamic process. It lacks an evaluation of the dynamic changes in Alu methylation over time in individual follow-ups, making it impossible to determine its temporal progression.

The article does not incorporate cell or animal experiments to verify the impact of Alu methylation on gene expression. The regulatory relationship between Alu methylation and key gastric cancer-related genes (such as CDH1, RASSF1A, and MLH1) remains unexplored.

The study uses the COBRA (Combined Bisulfite Restriction Analysis) method, which can only detect a limited number of CpG sites, making it inaccurate for quantifying genome-wide methylation levels. The lack of higher-resolution detection methods (such as BSP, qMSP, or MassARRAY) may lead to partial loss of methylation information. Additionally, Alu elements exist in different subtypes (e.g., Alu J, Alu S, and Alu Y), which may have distinct methylation patterns. The study does not classify specific Alu subtypes, potentially affecting data interpretation.

The study does not assess whether DNA demethylases (TET family) are involved in Alu methylation regulation and fails to explore the impact of reversible epigenetic regulation on gastric cancer.

Experimental design

See above

Validity of the findings

See above

Additional comments

See above

---

## Round 0.2 · Major Revisions

Thank you for the addressing the reviewers concerns. Before sending the new version to them, it is necessary to address a major concern of the editor. In the original version, in lines 201-203 it was written that it was PCR products which are treated with sodium bisulfite and this led to the question about the method.
In the revised version the same sentence is found in lines 236-238, meaning that the previous concern remains. In general extracted DNA is treated with bisulfite prior any additional step such as PCR. It had been suggested to provide proof of principle that the method used is really able to provide reliable information by applying it on commercial available methylated vs unmethylated DNA.

It is absolutely necessary to clarify this issue before the manuscript can be processed further.

---

## Round 0.3 · accepted · Accept

One reviewer is late in reporting however, the authors have addressed the two other reviewers comments in a satisfactory manner, therefore the decision is made to proceed. The manuscript is ready for publication.

Reviewer 2 ·

Basic reporting

no comment

Experimental design

no comment

Validity of the findings

no comment

Additional comments

no comment

Reviewer 3 ·

Basic reporting

no comment

Experimental design

no comment

Validity of the findings

no comment

Additional comments

no comment